# High epilepsy prevalence and excess mortality in onchocerciasis-endemic counties of South Sudan: A call for integrated interventions

Luís-Jorge Amaral[1,2]*, Stephen Raimon Jada[3], Jane Y. Carter[4], Yak Yak Bol[5], María-Gloria Basáñez[2], Charles R. Newton[6,7], Joseph N. Siewe Fodjo[1], Robert Colebunders[1,8]

**1** Global Health Institute, University of Antwerp, Antwerp, Belgium, **2** UK Medical Research Council Centre for Global Infectious Disease Analysis, and London Centre for Neglected Tropical Disease Research, Department of Infectious Disease Epidemiology, School of Public Health, Imperial College London, London, United Kingdom, **3** Amref Health Africa, Juba, South Sudan, **4** Amref Health Africa Headquarters, Nairobi, Kenya, **5** Neglected Tropical Diseases Unit, Ministry of Health, Juba, South Sudan, **6** Department of Psychiatry, University of Oxford, Oxford, United Kingdom, **7** Neurosciences Unit, Clinical Department, KEMRI-Wellcome Trust Research Programme-Coast, Kilifi, Kenya, **8** Department of Tropical Biology, Liverpool School of Tropical Medicine, Liverpool, United Kingdom

* luis-jorge.telesdemenesesdoamaral@uantwerpen.be

## Abstract

### Background

Epilepsy is a major health concern in onchocerciasis-endemic regions with intense transmission, where the infection is associated with a high epilepsy burden. This study investigated epilepsy prevalence and mortality in five onchocerciasis-endemic counties of South Sudan, and the association between onchocerciasis transmission and epilepsy, including probable nodding syndrome (pNS).

### Methodology

House-to-house cross-sectional surveys (2021–2024) identified persons with suspected epilepsy (sPWE) and retrospectively documented deaths among sPWE and individuals without epilepsy (IWE). Epilepsy diagnoses, including pNS, were confirmed by trained clinicians. Ongoing transmission was assessed using anti-Ov16 seroprevalence in children aged 3–9 years. Age- and sex-standardised epilepsy, pNS and anti-Ov16 prevalence were calculated, along with age- and sex-standardised mortality rates and standardised mortality ratios (SMRs) with 95% confidence intervals (95%CIs), using IWE as the reference population. Weighted arcsin-transformed linear regression was used to explore the association between epilepsy and anti-Ov16 prevalence.

**Data availability statement:** All relevant data are within the paper and its Supporting Information files.

**Funding:** The study was funded by an R2HC grant (Project ID: 78791403 to SRJ) and a grant from the Italian Agency for Development Cooperation (Project Code: AID011898 to SRJ) to Amref Health Africa, and the European Research Council (ERC 671055 to RC) to RC. L-JA received funding from the La Caixa Foundation (grant number B005782). JNSF received funding from the Research Foundation – Flanders (FWO, grant number 1296723N). M.G.B. acknowledges funding from the MRC Centre for Global Infectious Disease Analysis (MR/X020258/1), funded by the UK Medical Research Council (MRC). This UK-funded award is carried out in the frame of the Global Health EDCTP3 Joint Undertaking. The funders had no role in study design, data collection and analysis, decision to publish, or preparation of the manuscript.

**Competing interests:** The authors have declared that no competing interests exist.

## Principal findings

Among 34,019 individuals screened, 166 deaths occurred in 3,101 person-years for sPWE versus 466 deaths in 63,420 person-years for IWE. Epilepsy prevalence was 4.1% (range: 2.3-7.1%), and pNS prevalence was 1.5% (range: 0.6-2.2%). Anti-Ov16 seroprevalence among children was 23.3% (range: 1.4-44.1%). Each 1.0 percentage point increase in standardised anti-Ov16 seroprevalence was statistically significantly associated with an average rise of 0.10 percentage points in standardised epilepsy prevalence and 0.04 percentage points in standardised pNS prevalence. Median age at death was lower for sPWE (20 years) than IWE (38 years; Mann-Whitney U-test $p$-value $< 0.0001$). Standardised mortality rates per 1,000 person-years were statistically significantly higher in sPWE (67.6, 95%CI: 52.6-87.1) than in IWE (9.0, 95%CI: 7.8-10.3). The overall SMR was 6.9 (95%CI: 5.9-8.0), indicating sPWE were seven times more likely to die than IWE.

## Significance

The high epilepsy burden in onchocerciasis-endemic areas is driven by elevated epilepsy prevalence and mortality. Integrated onchocerciasis and epilepsy programmes must be strengthened to decrease epilepsy incidence and ensure uninterrupted access to antiseizure medication.

---

### Author summary

Epilepsy is a serious neurological condition that affects millions of people worldwide. In parts of sub-Saharan Africa where river blindness (onchocerciasis) is common, epilepsy prevalence is often high, and many individuals develop a severe form called nodding syndrome. In this study, we conducted house-to-house surveys in five counties of South Sudan, in a region where onchocerciasis remains widespread. We found that epilepsy prevalence was 4%—more than twice the regional average—and people with epilepsy were nearly seven times more likely to die prematurely than those without epilepsy. The highest epilepsy prevalence (up to 7%) was observed in villages near fast-flowing rivers, where blackflies—the insects that transmit onchocerciasis—breed. In these same areas, we found that young children had high levels of antibodies against the parasite that causes onchocerciasis (up to 44%), suggesting ongoing transmission despite control efforts. These findings add to the growing evidence that ongoing onchocerciasis transmission may contribute to epilepsy burden. Our results highlight the urgent need to strengthen onchocerciasis elimination programmes and improve access to antiseizure medication for affected communities to potentially reduce new epilepsy cases and prevent epilepsy-related deaths.

## 1. Introduction

Neurological disorders are a leading cause of disease burden and premature mortality [1]. Epilepsy is a chronic neurological condition characterised by recurrent, unprovoked seizures and accounts for approximately 0.5% of the global disease burden [2,3]. Around 80% of persons with epilepsy (PWE) reside in low- and middle-income countries (LMICs) [4], where limited diagnostic, treatment and prevention resources exacerbate the epilepsy burden [5]. Although antiseizure medication (ASM) can effectively manage epilepsy in over two-thirds of PWE with timely and consistent administration [6], access to ASMs and adequate care in LMICs remains low [4].

The epilepsy treatment gap is large in sub-Saharan Africa (SSA), where about 70% of PWE lack adequate care with consistent ASM treatment [5]. This gap is a major public health concern globally, and is further compounded in populations with high epilepsy prevalence, such as in highly endemic onchocerciasis areas [7–10]. Onchocerciasis, a parasitic disease caused by the filarial worm *Onchocerca volvulus* and transmitted by blackfly (*Simulium* spp.) vectors, was estimated to affect 20 million people in 2021 [3], with over 99% of infections in SSA [3]. Blackflies breed in fast-flowing water, increasing infection risk in nearby communities [11]. While onchocerciasis is typically characterised by ocular and cutaneous clinical manifestations, there is increasing evidence of its association with epilepsy, including nodding syndrome—an epilepsy encephalopathy [7,8,10,12–14]. Onchocerciasis-endemic areas are predominantly rural, impoverished and have limited medical infrastructure, complicating healthcare delivery [15].

Epilepsy has many causes, yet in over 40% of cases, its origin remains unknown [16]. Onchocerciasis-associated epilepsy (OAE) is one such example, for which two cohort studies in Cameroon showed that early childhood infection with high *O. volvulus* parasitic (microfilarial) loads was associated with an increased risk of developing epilepsy later in life [10,14]. In 2015, an estimated 400,000 individuals had OAE across onchocerciasis-endemic settings in Central and East Africa [17]. Addressing epilepsy in SSA is critical, as this region bears the highest global epilepsy burden [1,4], caused by high incidence of perinatal complications, central nervous system infections and head injuries, and exacerbated by factors including underdiagnosis, inadequate treatment, social stigma and misconceptions [18,19].

Mortality among PWE is estimated to be two to three times higher than in the general population [20,21], with even greater disparity in LMICs owing to unmet healthcare needs and inadequate seizure management [21]. A 2016 systematic review found that in LMICs, the primary causes of death among PWE were directly or indirectly related to epilepsy [21]. This high mortality among PWE is pronounced in SSA [21], including onchocerciasis-endemic areas [8,12]. A recent meta-analysis revealed statistically significantly higher mortality rates among PWE, including nodding syndrome, in onchocerciasis foci compared to PWE in non- to low-endemic regions [22].

Quantifying mortality in these areas will help fully assess the epilepsy burden in onchocerciasis-endemic regions and inform targeted public health interventions. This study investigated mortality rates among persons with suspected epilepsy (sPWE) compared to individuals without epilepsy (IWE) in five onchocerciasis-endemic counties of South Sudan and explored the association between ongoing *O. volvulus* transmission (anti-Ov16 seroprevalence in children) and epilepsy, including probable nodding syndrome (pNS).

## 2. Methods

### Ethics statement

Ethics approval was obtained from the South Sudan Ministry of Health (Maridi/Mundri: MOH/ERB3/2018; Mvolo: MOH/ERB50/2019; Wulu: MOH/RERP/P/35/15/05/2023-MOH/RERP/A/35/2023) and the Antwerp University Hospital, Belgium (Maridi/Mvolo/Mundri/Wulu: B300201940004). Personal information was encoded and treated confidentially. The participants included in this manuscript have given written informed consent to publication of their case details.

**Informed consent statement**

The aims and procedures of the study were explained to all participants in the language of their choice. Written informed consent was obtained from each participant, either through their signature or thumbprint. For minors and individuals unable to provide consent themselves, written consent was obtained from a parent or caregiver. Additionally, assent was obtained from adolescents (aged 12–18 years) and individuals capable of understanding but unable to provide full consent.

## 2.1  Study design and settings

This study employed a two-step, cross-sectional, door-to-door survey methodology across five onchocerciasis-endemic counties in South Sudan to identify PWE and retrospectively document deaths among sPWE and IWE (Fig 1). Surveys were conducted in Maridi (9–19th March 2022), Mvolo (4–18th June 2022), Mundri East (26th June–8th July 2021), Mundri West (26th June–8th 2021) and Wulu (2nd–5th February 2024). Previously published findings from these surveys reported epilepsy prevalence ranging from 3.3% (Mundri) to 4.5% (Mvolo), and incidence rates per 100,000 person-years ranging from 77 (Mundri) to over 300 (Maridi, Mvolo) [8,12,23]. Most PWE met the epidemiological criteria for OAE: onset between ages 3 and 18 years, living for ≥3 years in an onchocerciasis-endemic area, normal psychomotor development prior to epilepsy onset, and no obvious cause of epilepsy [24]. This applied to approximately 80% in Maridi, Mvolo and Mundri and 60% in Wulu [8,12,13,23]. Among children aged 3–9 years, anti-Ov16 seroprevalence ranged from 15.1% in Wulu to 30.7% in Maridi [8,12,13,23,25].

The selected counties were classified as meso- to hyperendemic for onchocerciasis (≥40% microfilarial prevalence) prior to control [26,27], with Mvolo reaching holoendemicity (90% prevalence) [8]. All five counties are endemic to nodding syndrome [13,28,29]. To control onchocerciasis, annual community-directed treatment with ivermectin (CDTI) has been introduced gradually in South Sudan since 2002 [8]. Recent surveys indicated that CDTI coverage in the study counties has remained suboptimal (below the minimal 65–80% coverage threshold of the total population at-risk recommended for *O. volvulus* elimination) [8,12,13,23,30]. Given this low coverage and the high OAE and anti-Ov16 prevalence in children,

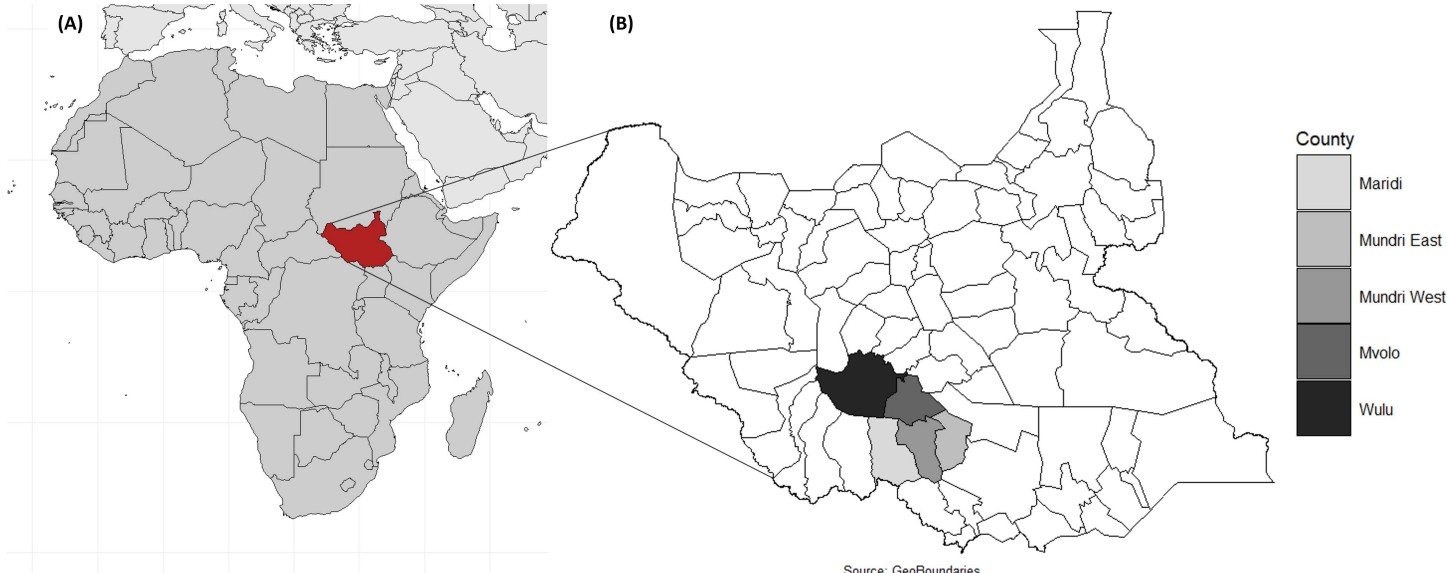

**Fig 1. Location of (A) South Sudan and (B) study counties (Maridi, Mundri East, Mundri West, Mvolo and Wulu).** Base maps sourced from (A) Natural Earth, https://www.naturalearthdata.com, and (b) GeoBoundaries, https://www.geoboundaries.org.

the South Sudan Ministry of Health increased CDTI frequency to twice a year in 2023 (excepting Wulu, where OAE had not yet been investigated) [29]. In Maridi, control interventions have already been intensified, including a "Slash & Clear" vector control strategy to reduce blackfly populations since 2019 and biannual CDTI since 2021 [12].

## 2.2 Sampling and participants

A multi-stage sampling approach was used to identify PWE and retrospectively document deaths among sPWE and IWE. Within each county, villages were prioritised according to their proximity to rivers, as epilepsy prevalence has been observed to decrease with increasing distance from blackfly breeding grounds [13,31,32].

Study villages were grouped based on their proximity to blackfly breeding grounds into high-risk sites (located within ≤2 kilometres) and low-risk sites (>2 kilometres) [33,34]. Table 1 details these classifications by county. The door-to-door approach included all residents within the study villages during the survey period.

## 2.3 Data collection and study procedure

Research assistants administered questionnaires to all household members to gather sociodemographic information and identify sPWE using a validated 5-question screening tool adapted from previous studies (Supplementary, Questionnaires A—D in S1 Appendix) [36,37]. A positive response to any screening question identified sPWE. The questions were translated from English to Arabic by local team members, then back-translated into English to ensure no loss of meaning, and pilot-tested in ten local households.

**Table 1. Classification of study villages according to their distance from rivers with confirmed vector (*Simulium damnosum* sensu lato) breeding grounds for the five South Sudanese study counties.**

| County | River | Payam (villages) | | Observations |
|---|---|---|---|---|
| | | High-risk site (≤ 2 km from breeding ground/river) | Low-risk site (> 2 km from breeding ground/river) | |
| Maridi | Maridi River | • Maridi (Kazana-1, Kazana-2, Kwanga and Hai Matara) | • Maridi (Hai Gabat, Hai Tarawa, Mudubai and Nagbaka) | Maridi Dam is the only known blackfly breeding ground in the area [32]. |
| Mundri East | Yei River Kembe River | – | • Lozoh (Lui Town) | Yei River is confirmed to have blackfly breeding grounds [23,27]. |
| Mundri West | Yei River | • Amadi (Hai Gabat) • Mundri Centre (Hai Lenderwa and Hai Ngulawa) | • Amadi (Hai Malakia) • Mundri Centre (Hai Facki and Hai Mirikalanga) | The research team observed blackfly breeding grounds in Mundri Centre (near Hai Lenderwa) and Amadi (near Hai Gabat) (Dr Stephen R. Jada, personal communication). |
| Mvolo[a] | Naam River | • Bagori (Hai Gira) • Mvolo (Dogoyabolu, Domeri, Dotriba, Hai Delib, Hai Dukaburo, Hai Korosona, Hai Masura, Hai Matara, Hai Muduria, Kila, Kperi, Korbabang, Mbadumu, Midikanunu, Minikolome, Muaskar and Tiboro) • Yeri (Dawanza, Dogabi, Dogereng, Hai Bogori, Hai Diocese, Hai Malakia, Hai Zira, Jebel Mari, Kelebala, Lodogo, Winikasa and Yeri Center) | – | Naam River contains multiple blackfly breeding grounds [35]. |
| Wulu | Naam River | • Domoloto (Woko Village) | • Domoloto (Tonjo) • Makundi (Kombi, Makundi Centre and War-Pac) | |

[a]Some villages in Mvolo may be slightly more than 2 kilometres from the breeding grounds. However, due to Mvolo's historical and current intense blackfly biting and onchocerciasis holoendemicity [8,35], all villages surveyed in this county were assumed as high risk for blackfly exposure.

Living participants who screened positive for epilepsy on the questions (sPWE) were referred to a clinician trained in epilepsy diagnosis. The clinician collected a detailed medical history and performed a clinical examination to confirm or exclude the diagnosis (Supplementary, Questionnaire E in S1 Appendix) using the 2014 International League Against Epilepsy (ILAE) criteria, which define epilepsy as the occurrence of two or more unprovoked seizures at least 24 hours apart [2]. Adherence to ASM was defined as self-reported daily intake during the previous week.

Confirmed PWE were further assessed for pNS, defined as a history of repeated, involuntary head-dropping seizure episodes during brief periods of reduced consciousness, with onset between the ages of three and 18 years. In addition, at least one of the following had to be present: other neurological impairments, spatial clustering of cases, stunted growth, physical deformities, or delayed/absent secondary sexual characteristics [38].

Household heads were also asked if any deaths had occurred in the household within the previous 24 months. For each death, respondents indicated whether the deceased had been known to have epilepsy (one of the screening questions), along with details of sex and age at death. Deceased individuals known to have epilepsy were classified as sPWE; otherwise, they were considered IWE.

To assess ongoing *O. volvulus* transmission, the seroprevalence of IgG4 antibodies to the Ov16 antigen (anti-Ov16 seroprevalence) in children aged 3–9 years was measured. Blood samples were collected via finger prick and tested using the Ov16 rapid diagnostic test (RDT, SD Bioline, Inc., Gyeonggi-do, South Korea). Results were read at 30 minutes and performed in the field following the manufacturer's instructions.

Households were defined as all individuals residing together and sharing meals. Residency status was categorised as "native" (household head born and raised in, or residing in the village for ≥20 years) or "immigrant" (otherwise).

## 2.4  Statistical analysis

Continuous variables were inspected for normality using histograms and the Shapiro-Wilk test. Non-normally (non-Gaussian) distributed data were summarised as medians with interquartile ranges (IQRs, 1st quartile to 3rd quartile). Categorical variables were analysed as frequencies and proportions (expressed as percentages). To explore differences in sociodemographic variables between sites, non-parametric tests were employed and p-values (p) and effect sizes determined (detailed methodology and results in Text A in S1 Appendix, Tables A and B in S1 Appendix).

County-level crude prevalence estimates for lifetime epilepsy, pNS and anti-Ov16 seropositivity in children aged 3–9 years were published previously [8,12,13,23]. For this study, these data were further stratified by high- and low-risk sites (Table 1) and standardised. Epilepsy, including pNS, prevalence was standardised by sex and two age groups: 0–20 and >20 years, based on the typical OAE onset time and the available sample size per study site. Similarly, anti-Ov16 seroprevalence was standardised by sex and two age groups: 3–6 and 7–9 years, based on onchocerciasis cumulative exposure over time. Standardisation used the internal (surveyed) population as the reference. Prevalence values were expressed as percentages, alongside 95% Wilson score confidence intervals (95% CIs) with continuity correction [39].

Weighted linear regression examined the association between age and sex-standardised anti-Ov16 seroprevalence and age- and sex-standardised epilepsy prevalence per site, with the regression line and 95% CIs illustrated in a scatterplot. Before fitting the regression model, epilepsy prevalence was arcsine-transformed ($y = \sin^{-1}\sqrt{\frac{epilepsy\ prevalence}{100}}$) to stabilise variance and accommodate its proportional nature [40]. Weights were calculated based on the inverse of the total variance of each prevalence and seroprevalence estimate. This process was repeated to analyse the association between age- and sex-standardised anti-Ov16 seroprevalence and age- and sex-standardised pNS prevalence.

Due to the relatively low frequency of mortality events, the mortality analysis was conducted at the county level to maximise statistical power. Mundri East and West Counties were combined as they were part of the same, smaller survey. Mortality rates, expressed per 1,000 person-years, were calculated by dividing the number of deaths recorded in each county for both sPWE and IWE by the respective total person-years for each group. To adjust for age- and sex-specific differences, mortality rates were directly standardised using the combined IWE population as the standard population.

For mortality standardisation, age groups were divided into 10-year increments, combining groups with fewer than five deaths and capped at 50 years of age, as only 1% of sPWE exceeded this age. For each age and sex group per county, expected deaths were calculated by applying age- and sex-specific mortality rates from the overall standard population to the corresponding population size of each group. Standardised mortality risk ratios (relative risk) were calculated by dividing standardised mortality rates among sPWE by those among IWE, with 95% CIs obtained using the log-normal approximation based on the confidence bounds of the directly standardised mortality rates.

Age- and sex-standardised mortality ratios (SMRs) were calculated as follows:

$$\text{SMR} = \frac{\text{Observed deaths in sPWE}}{\text{Expected deaths in sPWE (using the respective IWE mortality rate as a standard)}} \quad (1)$$

To evaluate the accuracy of identifying epilepsy status among deceased individuals through retrospective household interviews, positive predictive values (PPVs) were calculated using data from living sPWE. These individuals were identified using the same screening question: "Has it ever been said that he/she is epileptic or has experienced two or more seizures?". The PPV was determined as:

$$\text{PPV} = \frac{\text{Confirmed living individuals with epilepsy (initially identified as sPWE by the screening question)}}{\text{Total living individuals identified as sPWE by the screening question}} \quad (2)$$

## 3. Results

### 3.1 Sociodemographic characteristics of surveyed households

A total of 5,229 households, comprising 34,051 individuals, were surveyed across the five counties and their eight high- and low-risk sites (Table A in S1 Appendix). Household size was generally consistent across sites, with a median of 6 (IQR: 5–8) individuals. The proportion of household heads native to their current village ranged from 72% in Wulu's low-risk site to 96% in Wulu's high-risk site, with larger sites at around 80–90%. Among non-native heads, the median residence duration was 5 (IQR: 3–9) years, with negligible variation across sites.

Farming was the primary occupation, involving >80% of households in all sites. Fishing was mostly reported in Mvolo's (33.3%), Wulu's (30.0%) and Mundri West's (21.3%) high-risk sites, and Mundri West's low-risk site (16.8%). Cattle-rearing was most frequent in Mvolo (14.2%), compared to 1–7% in other sites. Pig-rearing was uncommon, ranging from 0.8% in Mvolo to 4.6% in Mundri West's high-risk site. Other employment activities, including craftsmanship and government work, were reported in 30–40% of households across all sites excepting Wulu, a more remote area where engagement in these activities was lower (6–8%). Detailed site-specific household characteristics are available in Table A in S1 Appendix.

### 3.2 Sociodemographic characteristics of surveyed individuals

Among the 34,051 individuals surveyed, the median age was 17 (IQR: 8–29) years, with negligible variation across sites (Table B in S1 Appendix). The proportion of female participants was consistent across sites, with an overall percentage of 51.6% (range: 50.7–53.8%). Ivermectin intake in the year preceding each county-specific survey varied substantially by location, with most sites having at least half of individuals reporting taking ivermectin (49–74%), except Mundri East (34%) and Mvolo (24%). Detailed site-specific individual characteristics are provided in Table B in S1 Appendix.

### 3.3 Anti-Ov16 seroprevalence among children and overall epilepsy prevalence

Age- and sex-standardised anti-Ov16 seroprevalence among children aged 3–9 years was 22.3% (95% CI: 19.6–25.2%) overall (Table C in S1 Appendix). Seroprevalence was highest in Maridi's high-risk site (38.3%, 95% CI: 32.2–44.9%)

followed by Wulu's (32.2%, 95% CI: 20.6–46.2%), Mvolo's (26.7%, 95% CI: 20.0–34.6%) and Mundri West's (16.1%, 95% CI: 10.3–24.1%) high-risk sites. Lower seroprevalence was recorded in Maridi's (11.5%, 95% CI: 7.1–17.9%), Mundri West's (4.4%, 95% CI: 1.2–12.8%) and Wulu's (1.7%, 95% CI: 0.1–9.4%) low-risk sites. Data were not collected for Mundri East.

Lifetime epilepsy prevalence was 4.1% (95% CI: 3.9–4.3%), and pNS prevalence was 1.5% (95% CI: 1.4–1.7%) (Table 2). The prevalence of suspected lifetime epilepsy, including both confirmed and unconfirmed or rejected cases, was 4.3% (95% CI: 4.1–4.5%) and is provided in Table D in S1 Appendix. Overall, 53.4% (725/1,357) of PWE reported adherence to ASM. Adherence was high in Maridi (86.6%, 496/573) and Mundri East and West (88.1%, 74/84), but markedly lower in Mvolo (23.1%, 149/645) and Wulu (10.9%, 6/55).

Age- and sex-standardised lifetime epilepsy prevalence was highest in Wulu's (7.1%, 95% CI: 5.0–10.0%) and Maridi's (5.1%, 95% CI: 4.6–5.6%) high-risk sites. Both were statistically significantly greater than in their counterpart sites located further from blackfly breeding grounds: Wulu's (2.5%, 95% CI: 1.6–3.8%) and Maridi's (2.7%, 95% CI: 2.4–3.2%) low-risk sites. Mvolo (high-risk site only) also had a high epilepsy prevalence (4.3%, 95% CI: 4.0–4.7%). Similarly, Mundri West's high-risk site had a higher epilepsy prevalence (4.5%, 95% CI: 3.2–6.3%) than its low-risk site (2.3%, 95% CI: 1.5–3.5%) (p = 0.016). Mundri East (low-risk site only) had a prevalence (3.0%, 95% CI: 2.0–4.4%) comparable to the other low-risk sites.

Higher age- and sex-standardised pNS prevalence was recorded in areas closer to blackfly breeding grounds, specifically in Maridi's (2.1%, 95% CI: 1.8–2.5%), Mvolo's (1.8%, 95% CI: 1.6–2.0%), Mundri West's (1.5%, 95% CI: 0.8–2.8%) and Wulu's (1.1%, 95% CI: 0.4–2.7%) high-risk sites. In contrast, pNS prevalence was below 1.0% in low-risk sites, although cases were still observed.

**Table 2. Epilepsy, including probable nodding syndrome (pNS), and pNS prevalence per study county in high- and low-risk sites in South Sudan.**

| County (year of survey) | Classification by distance to vector breeding grounds | Confirmed epilepsy | | Probable nodding syndrome (pNS) | |
|---|---|---|---|---|---|
| | | Number; Total screened | Age- and sex-standardised prevalence % (95% CI) | Number; Total screened | Age- and sex-standardised prevalence % (95% CI) |
| Maridi (2022) | High-risk | 390; 7,679 | 5.1 (4.6–5.6) | 162; 7,679 | 2.1 (1.8–2.5) |
| | Low-risk | 183; 6,702 | 2.7 (2.4–3.2) | 40; 6,702 | 0.6 (0.4–0.8) |
| | Overall | 573; 14,381 | 4.0 (3.7–4.3) | 202; 14,381 | 1.4 (1.2–1.6) |
| Mundri East (2021) | Low-risk only | 26; 859 | 3.0 (2.0–4.4) | 6; 859 | 0.7 (0.3–1.6) |
| Mundri West[b] (2021) | High-risk | 35; 764 | 4.5 (3.2–6.3) | 12; 764 | 1.5 (0.8–2.8) |
| | Low-risk | 23; 960 | 2.3 (1.5–3.5) | 5; 960 | 0.5 (0.2–1.3) |
| | Overall | 58; 1,724 | 3.3 (2.5–4.2) | 17; 1,724 | 1.0 (0.6–1.6) |
| Mvolo (2022) | High-risk only | 650; 15,070 | 4.3 (4.0–4.7) | 273; 15,070 | 1.8 (1.6–2.0) |
| Wulu (2024) | High-risk | 33; 461 | 7.1 (5.0–10.0) | 5; 461 | 1.1 (0.4–2.7) |
| | Low-risk | 22; 892 | 2.5 (1.6–3.8) | 6; 892 | 0.7 (0.3–1.6) |
| | Overall | 55; 1,353 | 4.1 (3.1–5.3) | 11; 1,353 | 0.8 (0.4–1.5) |
| Total | | 1,362; 33,387 | 4.1 (3.9–4.3) | 509; 33,386 | 1.5 (1.4–1.7) |

*Note*: In some living persons, suspected epilepsy was not confirmed and these were removed from the calculations of confirmed epilepsy and probable nodding syndrome: 4 in Maridi (3 in the high-risk site and 1 in the low-risk site); 3 in Mundri East; 1 in Mundri West (high-risk site); 22 in Mvolo; 2 in Wulu County (1 in each of the high-risk and the low-risk sites).

Abbreviations: CI, confidence interval; N, number; T, total screened.

The PPV of the epilepsy screening question—whether the sPWE had been known to have epilepsy or experienced ≥2 seizures—was 97.7% overall (95% CI: 96.6–98.5%) and consistent across all counties (97–100%) excepting Wulu (62.3%, 95% CI: 47.9–74.9%) (Table E in S1 Appendix).

Weighted linear regression revealed statistically significant positive associations between age- and sex-standardised anti-Ov16 seroprevalence in children (3–9 years) and both age- and sex-standardised epilepsy prevalence (Table 3; Fig 2A) and pNS prevalence (Table 3; Fig 2B). For practical interpretation, within the observed range of anti-Ov16 seroprevalence, a one percentage point increase (e.g., from 20% to 21%) is estimated to correspond to an average increase of 0.10 (range: 0.08–0.12) percentage points in epilepsy prevalence and 0.04 (range: 0.03–0.05) percentage points in pNS prevalence (see Text B in S1 Appendix for detailed calculations). The models explained most of the variation in prevalence (adjusted R² values of 0.75 for epilepsy and 0.67 for pNS; Table 3). Both models had significant intercepts, indicating baseline prevalence is approximately 2.3% (95% CI: 1.5–3.2%) for epilepsy and 0.5% (95% CI: 0.3–0.9%) for pNS when anti-Ov16 seroprevalence is zero.

### 3.4 Mortality rates and standardised mortality ratios

The overall median age at death was statistically significantly lower for sPWE (20 years, IQR: 15–27) than for IWE (38 years, IQR: 18–58) (Mann-Whitney U test, p<0.0001) (Table 4). This trend was observed in all counties excepting Wulu, where the median age at death IWE was lower in IWE (20 years; IQR: 3–35) compared to sPWE (24 years, IQR: 18–25; p=0.64). This deviation in Wulu was due to a higher proportion of deaths of early childhood deaths (aged ≤5 years) among IWE (42%, 11/26 deaths) compared to sPWE (11%, 1/9 deaths).

Crude mortality rates were consistently and statistically significantly higher among sPWE compared to IWE in all counties. Age- and sex-standardised mortality rates further highlighted this disparity. In Maridi, the standardised mortality rate for sPWE was 41.8 per 1,000 person-years (95% CI: 31.1–55.3) compared to 9.8 (95% CI: 8.6–11.0) for IWE, yielding a standardised mortality risk ratio (RR) of 4.3 (95% CI: 3.1–5.8) and a standardised mortality ratio (SMR) of 5.2 (95% CI: 3.9–6.7). In Mundri East and West, sPWE had an extremely high standardised mortality rate of 79.7 per 1,000 person-years (95% CI: 41.0–142.0), statistically significantly greater than the 8.1 (95% CI: 5.9–10.9) observed in IWE, corresponding to an RR of 9.8 (95% CI: 4.9–19.7) and an SMR of 6.2 (95% CI: 3.5–10.3).

In Mvolo, the disparity was even more pronounced, with sPWE mortality at 94.2 per 1,000 person-years (95% CI: 70.8–126.0), versus 7.8 (95% CI: 6.8–9.0) for IWE. This resulted in an RR of 12.1 (95% CI: 8.8–16.6) and an SMR of 9.3 (95% CI: 7.6–11.3). Wulu also demonstrated considerably elevated mortality among sPWE, with a standardised mortality rate of 54.3 per 1,000 person-years (95% CI: 24.7–104.0) compared to 10.2 (95% CI: 6.7–14.9) for IWE, yielding an RR of 5.3 (95% CI: 2.3–12.1) and an SMR of 4.8 (95% CI: 2.2–9.1).

Overall, across all counties surveyed from 2021–2024, the standardised mortality rate among sPWE was statistically significantly higher at 67.6 per 1,000 person-years (95% CI: 52.6–87.1), compared to 9.0 (95% CI: 7.8–10.3) among IWE.

**Table 3. Results of weighted arcsine-transformed linear regression of the age-and sex-standardised epilepsy prevalence (y) and age- and sex-standardised anti-Ov16 seroprevalence ($x_1$, epilepsy model) and of age-and sex-standardised probable nodding syndrome prevalence (y) and age- and sex-standardised anti-Ov16 seroprevalence ($x_1$, pNS model).**

| Coefficients | Epilepsy model | | pNS model | |
|---|---|---|---|---|
| | Estimate (95% CI) | P-value | Estimate (95% CI) | P-value |
| Intercept ($\beta_0$) | 0.1513 (0.1229–0.1798) | <0.001 | 0.0740 (0.0501–0.0979) | <0.001 |
| Anti-Ov16 seroprevalence ($\beta_1$) | 0.0025 (0.0010–0.0039) | 0.008 | 0.0018 (0.0005–0.0030) | 0.015 |
| R² | 0.79 | – | 0.73 | – |
| Adjusted R² | 0.75 | – | 0.67 | – |

*Note*: The model is $y = \beta_0 + \beta_1 * x_1$

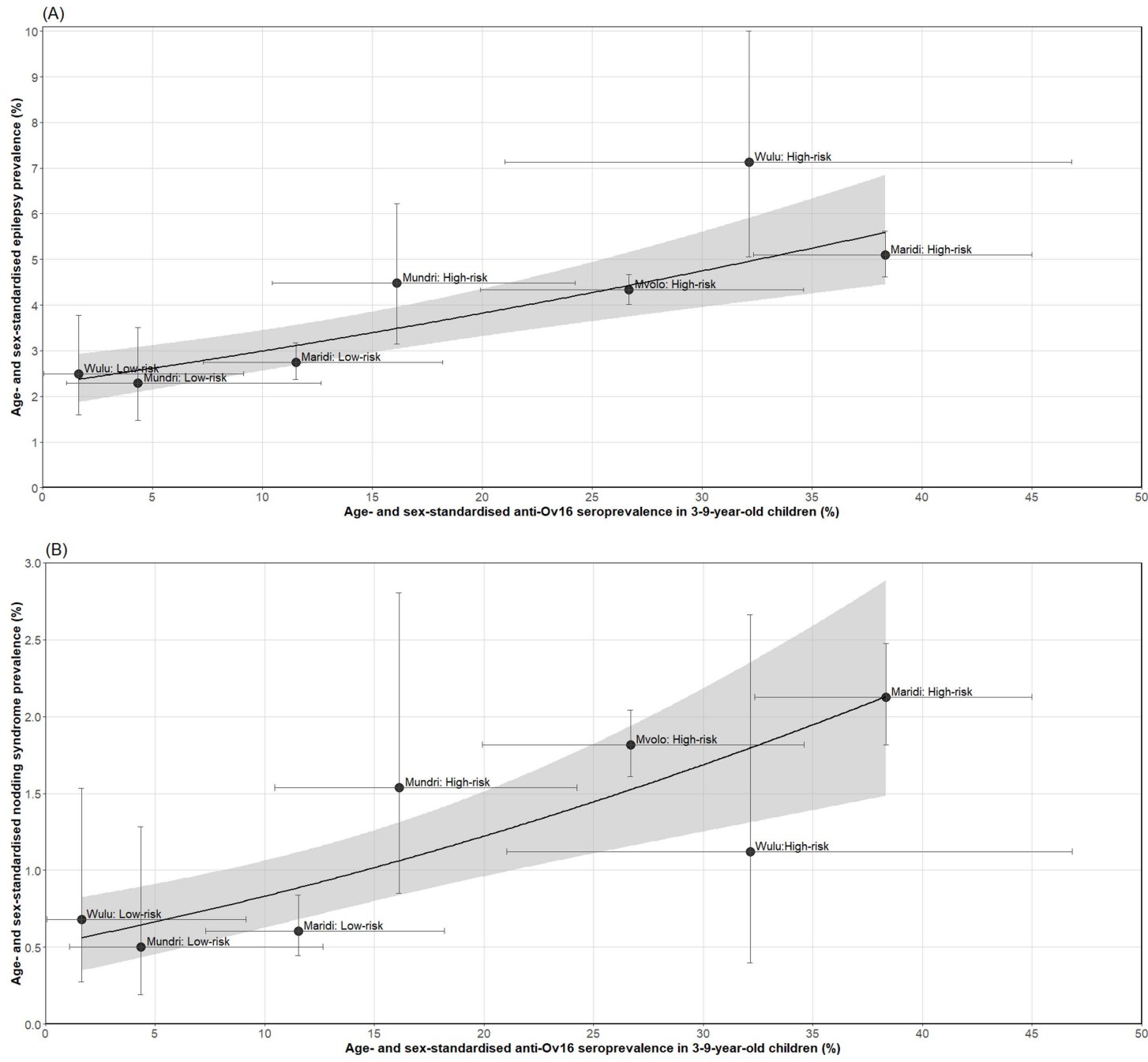

**Fig 2. Relationship between age- and sex-standardised anti-Ov16 seroprevalence in children aged 3–9 years and age- and sex-standardised confirmed prevalence of epilepsy (A) and probable nodding syndrome (B) across eight study sites in five counties of South Sudan.** Black circles represent study sites. Error bars are 95% confidence intervals (CIs) for anti-Ov16 seroprevalence (horizontal bars) and epilepsy or probable nodding syndrome prevalence (vertical bars). Grey-shaded areas are the 95% CIs around the weighted linear regression dark line. Mundri East was excluded from this analysis, as no anti-Ov16 testing was conducted in this location. *Note*: The regression lines and 95% CIs are based on arcsin-transformed regression outputs that have been back-transformed to the original scale for interpretability.

**Table 4. Summary of person-years, deaths, median age at death, crude mortality rates, age- and sex-standardised mortality rates (per 1,000 person-years), and standardised mortality ratios (SMR) for persons with suspected epilepsy (sPWE) and individuals without epilepsy (IWE) across counties, South Sudan.**

| County (year of survey) | Population group | Person-years | Number of deaths | Median age at death years (IQR) | Crude mortality rate Deaths per 1,000 person-years (95% CI) | Age- and sex-standardised mortality rate Deaths per 1,000 person-years (95% CI) | Age- and sex-standardised mortality risk ratio (RR) (95% CI) | Age- and sex-standardised mortality ratio (SMR) (95% CI) |
|---|---|---|---|---|---|---|---|---|
| Maridi (2022) | sPWE | 1,269 | 55 | 19 (15–25) | 43.3 (33.1–56.4) | 41.8 (31.1–55.3) | 4.3 (3.1–5.8) | 5.2 (3.9–6.7) |
| | IWE | 27,365 | 283 | 40 (23–63) | 10.3 (9.2–11.6) | 9.8 (8.6–11.0) | | |
| Mundri East & West (2021) | sPWE | 216 | 15 | 23 (16–30) | 69.4 (40.8–114) | 79.7 (41.0–142) | 9.8 (4.9–19.7) | 6.2 (3.5–10.3) |
| | IWE | 5,037 | 44 | 42 (22–60) | 8.7 (6.4–11.8) | 8.1 (5.9–10.9) | | |
| Mvolo (2022) | sPWE | 1,482 | 99 | 20 (15–28) | 66.8 (54.9–81.0) | 94.2 (70.8–126) | 12.1 (8.8–16.6) | 9.3 (7.6–11.3) |
| | IWE | 28,643 | 207 | 34 (16–50) | 7.2 (6.3–8.3) | 7.8 (6.8–9.0) | | |
| Wulu (2024) | sPWE | 159 | 9 | 24 (18–25) | 56.6 (27.9–108) | 54.3 (24.7–104) | 5.3 (2.3–12.1) | 4.8 (2.2–9.1) |
| | IWE | 2,560 | 26 | 20 (3–35) | 10.2 (6.8–15.1) | 10.2 (6.7–14.9) | | |
| Total (2021–2024) | sPWE | 3,101 | 166 | 20 (15–27) | 53.5 (46.0–62.2) | 67.6 (52.6–87.1) | 7.5 (5.6–10.0) | 6.9 (5.9–8.0) |
| | IWE | 63,420 | 466 | 38 (18–58) | 7.4 (6.7–8.1) | 9.0 (7.8–10.3) | | |

Abbreviations: IQR, interquartile range; CI, confidence interval; SMR, standardised mortality ratio; sPWE, persons with suspected epilepsy; IWE, individuals without epilepsy.

This resulted in a pooled RR of 7.5 (95% CI: 5.6–10.0), indicating that sPWE had nearly eight times the mortality rate of those without epilepsy. Similarly, the overall SMR of 6.9 (95% CI: 5.9–8.0) showed that observed deaths among sPWE were about seven times higher than expected.

## 4. Discussion

The current study underscores the substantial epilepsy prevalence and mortality among PWE in onchocerciasis-endemic areas of South Sudan. With a surveyed prevalence of 4.1% (95% CI: 3.9–4.3%), this value statistically significantly exceeds both the global average of 0.8% (95% CI: 0.6–0.9%) and the SSA average of 1.6% (95% CI: 1.2–2.0) [41]. This elevated prevalence confirms prior studies reporting an association between high *O. volvulus* transmission and increased epilepsy burden [7,8,10,12–14].

The high-risk sites of Maridi, Mundri West and Wulu Counties, located close to rivers with blackfly breeding grounds, exhibited statistically significantly higher prevalence of both epilepsy and anti-Ov16 seropositivity in children compared to their more distant counterparts (low-risk sites). This spatial trend was captured by a statistically significant weighted arcsin-transformed linear regression, with standardised epilepsy prevalence increasing by an average of 0.10 percentage points for each 1.0 percentage point rise in standardised anti-Ov16 seroprevalence in children. A statistically significant association was also found for standardised pNS prevalence, which rose by about 0.04 percentage points per 1.0 percentage point rise in standardised seroprevalence. These findings provide further evidence

for the link between onchocerciasis transmission and epilepsy, including pNS, within SSA and, specifically, in South Sudan [14,28].

An overall SMR of 6.9 (95% CI: 5.9–8.0) indicated that sPWE experienced nearly seven times more deaths than would be expected based on the age- and sex-specific mortality rates of IWE. This finding suggests a substantial excess mortality burden among sPWE and aligns with some of the highest SMRs reported in LMICs, as identified in a 2017 systematic review by the ILAE Task Force. These included SMRs of 6.3 in urban Ecuador, 6.5 in rural Kenya and 7.2 in an onchocerciasis-endemic area of West Uganda [21]. Comparable patterns have been documented in Mahenge, Tanzania, one of the most endemic areas for onchocerciasis in East Africa, although SMRs were not explicitly calculated [42]. Furthermore, a recent meta-analysis confirmed that PWE in SSA are about five times more likely to die than IWE (95% CI: 3.5–6.8), with the mortality risk ratio rising to over six-fold in high-risk onchocerciasis areas [22].

Direct comparisons of the high standardised mortality rates observed in our study (34–94 deaths per 1,000 person-years among sPWE) with previous research are constrained by variations in standardisation methods. Nonetheless, these rates lie within previously documented ranges, reflecting the substantial mortality burden associated with epilepsy in onchocerciasis-endemic regions [22,32,42,43].

The distinction between RRs and SMRs in our analysis merits clarification. Directly standardised RRs quantified how much higher the mortality rate is among sPWE compared to IWE [44], adjusting for age and sex using the overall sampled population as the external standard. Thus, RRs directly underscore the increased public health burden of epilepsy-related deaths. In contrast, SMRs indicate how many deaths occurred among sPWE relative to what would be expected based on the mortality patterns of the IWE population (at county-level or overall) [44]. This approach adjusts to the actual age-sex distribution of the epilepsy group, thereby emphasising the excess deaths that could potentially be prevented by targeted interventions. In our study, the RRs were generally higher than SMRs as sPWE were younger than IWE. As a result, the SMRs assign greater weight to younger age bands, where absolute mortality and mortality differences are typically smaller, thus leading to lower SMRs relative to RRs. Despite this distinction, both measures consistently demonstrate substantially elevated mortality among sPWE, highlighting the urgent need for targeted public health interventions, including improved epilepsy care and enhanced prevention strategies.

The statistically significantly lower median age at death for sPWE (20 years) compared to IWE (38 years) further emphasises the premature mortality faced by PWE in these areas. This discrepancy is especially concerning for South Sudan, where three-quarters of the population resides in rural settings and is both youthful and growing rapidly [29]. Combined with suboptimal onchocerciasis control in South Sudan [26,29], this young demographic profile may exacerbate the epilepsy burden, as OAE onset typically occurs in childhood or adolescence, increasing the likelihood of premature mortality and more years lived with disability [22]. Median ages of epilepsy onset are 8 (IQR: 4–12) years in Wulu [13], 9 (IQR: 6–13) years in Mvolo [8] and 10 (IQR: 6–15) years in Mundri East and West and Maridi [12,23].

Wulu County was an outlier in the mortality analysis, with a median age at death for IWE (20 years) somewhat lower than for sPWE (24 years), driven by high mortality among children aged ≤5 years in the IWE group (42% of deaths) compared to the sPWE group (11%). This discrepancy is due to early childhood malaria (4/11 deaths among children aged ≤5 years without epilepsy) and the epilepsy age of onset in late childhood [13].

Access to ASM remains limited in South Sudan, with most families unable to sustain continuous treatment [5,45,46]. This was reflected in our study, where only half of PWE reported taking ASM daily in the week prior to the survey. Lack of access and inconsistent adherence to epilepsy treatment are known contributors to high mortality, as highlighted in a study in Kenya, where PWE who did not adhere to ASM treatment had mortality rates three times higher than those who consistently adhered [47]. Among the counties surveyed in our study, only Maridi and Mundri East and West have established epilepsy treatment programmes [12,23], which is mirrored in their higher ASM adherence (86% and 88%, respectively) compared to Mvolo (23.1%) and Wulu (10.9%).

In 2018, before the establishment of the epilepsy clinic with free ASM provision in Maridi, a baseline study in the area reported a mortality rate of 59 deaths per 1,000 person-years (96/1,631) among sPWE [32], which is higher than the 43 per 1,000 person-years documented for Maridi in our study (Poisson test p-value = 0.07). This reduction may reflect the positive impact of the epilepsy clinics and strengthened onchocerciasis elimination efforts since 2019 [12], and is further supported by findings from a cohort study in Mahenge, Tanzania [48], where a community-based epilepsy care programme offering free ASM and adherence support alongside enhanced onchocerciasis control significantly reduced seizure frequency among PWE. Most deaths within the Mahenge cohort were epilepsy-related and among participants who did not adhere to treatment [48].

From 1990 to 2016, epilepsy-related mortality rates significantly decreased globally, with high- and middle-income countries driving this trend through better treatment access [4]. These advances highlight that increasing treatment availability in LMICs—which can cost as little as US$5 annually— could substantially reduce epilepsy-related mortality [4,6]. Expanding community-based ASM distribution programmes or integrating epilepsy treatment with strengthened onchocerciasis elimination efforts through community drug distributors could enhance treatment access in underserved areas [49]. Community engagement, addressing stigma through awareness campaigns and fostering partnerships with local non-governmental organisations, policy-makers and international agencies could help ensure the sustainability of these interventions [5,45].

CDTI coverage across the five counties surveyed was suboptimal, ranging from 24.1% in Mvolo to 66% in Wulu, limiting efforts to control onchocerciasis and OAE. While coverage fluctuated somewhat over recent years, including during the COVID-19 pandemic, it remained below the minimal 65–80% recommended for effective elimination throughout the study period [8,12,30]. Strengthening elimination programmes could be followed by a marked reduction in epilepsy incidence, as reported in Maridi and Mvolo [8,12], where sustained and enhanced onchocerciasis control interventions were followed by statistically significant decreases in incident epilepsy cases. Mathematical modelling suggests that achieving at least 80% CDTI coverage in the overall population would be required to reduce onchocerciasis endemicity to a level where OAE could be eliminated [50]. With the South Sudan Ministry of Health's initiative to increase CDTI frequency to biannually in four of the study counties, understanding and addressing the factors behind poor CDTI uptake will be essential to maximise its impact on onchocerciasis elimination and consequent decline in epilepsy incidence. Incorporating epilepsy outcomes into routine onchocerciasis surveillance and programme evaluations could refine public health strategies, inform future global burden estimates and reinforce the importance of a holistic, community-centred approach to disease control.

### 4.1 Limitations

A key limitation of the study is its retrospective design, introducing potential recall, and information biases due to reliance on verbal household reports for mortality data. While epilepsy diagnoses among deceased individuals could only be suspected retrospectively, the screening question used showed high validity among living sPWE who were subsequently clinically assessed, yielding a high PPV of 97.7% (95% CI: 96.6–98.5%). In Wulu, however, the PPV (66%) was statistically significantly lower, likely reflecting a less stringent application of the diagnostic criteria for suspected epilepsy, which resulted in more individuals with febrile seizures being referred to as suspected epilepsy cases (15% of sPWE in Wulu were reclassified accordingly). This over-referral in Wulu was further compounded by a small sample size and poor child healthcare (evidenced by high under-five mortality) that could allow childhood fevers to persist and trigger seizures [13].

Another limitation relates to the small number of deaths recorded in certain counties. Although the overall number of deaths included in the study was high, smaller sample sizes in Mundri and Wulu resulted in wider CIs for mortality indicators (e.g., SMR 95% CI: 3.5–10.3 for Mundri, 2.2–9.1 for Wulu) compared to counties with larger sample sizes (e.g., SMR 95% CI: 3.9–6.7 for Maridi, 7.6–11.3 for Mvolo). This variability underscores the need for caution in interpreting county-specific results, particularly where uncertainty is denoted by wide CIs.

Additionally, the lack of reliable data on ASM access and use warrants further investigation given its effect on epilepsy outcomes. Moreover, the study could not employ epilepsy gold-standard diagnostic methods such as electroencephalography (EEG) or magnetic resonance imaging (MRI). However, the clinical epidemiological diagnostic methods used have been recognised by ILAE [51] and have consistently yielded similar epilepsy prevalence estimates in repeated cross-sectional surveys of comparable areas [8,12]. For example, in Mvolo, the epilepsy prevalence among those older than 20 years was 63.7 per 1,000 persons (95% CI: 58.2–69.7) in 2020 and 63.8 per 1,000 persons (95% CI: 57.9–70.2) in 2022 [8].

Although a high proportion of PWE in our study areas meet the epidemiological criteria for OAE (80% in Maridi, Mvolo and Mundri; 60% in Wulu) [8,12,13,23], we acknowledge that epilepsy in these regions is heterogeneous, and specific aetiologies may contribute to the overall burden. The lack of advanced epilepsy diagnostic tools, such as brain imaging or molecular investigations, constrained our ability to identify specific epilepsy aetiologies. Neurocysticercosis-related lesions, for example, could have been detected via imaging, although *Taenia solium* infection is unlikely to be a major contributor to epilepsy in the study areas. Only 1.5% of households reported pig-rearing, and just 2.3% of sPWE (20/876) came from families who reported this activity. This proportion is extremely low and not statistically significantly different from that of individuals without epilepsy (1.5%, 264/17,168; chi-squared test, p = 0.11).

The cause of death for sPWE was not consistently documented across counties, and detailed suspected cause-of-death data were only collected in Wulu. Although the limited sample size precludes robust conclusions, most deaths among sPWE in Wulu were epilepsy-related, consistent with previous findings from onchocerciasis-endemic areas [21,22,43,52].

Lastly, while the weighted arcsine-transformed linear regression estimates were age- and sex-standardised and populations had similar sociodemographic profiles, the model did not account for additional confounders such as migration between high- and low-risk sites. Such migration may explain why the pNS model intercept approached but did not reach zero, despite the expectation that pNS prevalence is strongly dependent on onchocerciasis endemicity [9]. Furthermore, the model did not incorporate the Ov16 RDT's suboptimal sensitivity (60–80%), potentially underestimating true seroprevalence [53]. Model intercepts also lay outside the observed range of anti-Ov16 seroprevalence and thus should be interpreted with caution.

## 4.2 Recommendations

Despite these limitations, our findings urgently call for integrated public health interventions to prevent and treat epilepsy in onchocerciasis-endemic areas. Given the substantial excess mortality observed in our study area, these findings strongly support the implementation of decentralised, community-based epilepsy care in other highly endemic, hard-to-reach settings, as successfully demonstrated in Tanzania [48]. Future studies would benefit from combining prospective cohort designs, including robust, physician-certified verbal autopsies and health registries for systematically assessing causes of death [54,55], with targeted mechanistic investigations, such as neuro-imaging, electroencephalography and broader infectious disease screening panels, to explore alternative epilepsy aetiologies [52]. However, such approaches are currently constrained by limited resources, remote locations and ongoing conflicts in highly endemic areas. Recent findings identifying viruses within the *O. volvulus* virome that can elicit human immune responses warrant further investigation to determine if they may be linked to the potential pathophysiology underlying OAE [56]. In parallel, a re-estimation of the burden of epilepsy in onchocerciasis foci is a pressing need to inform appropriate resource allocation for effectively addressing the combined burden. We therefore urge future iterations of the Global Burden of Disease Study to include OAE and excess mortality in the estimations of disability-adjusted life years for onchocerciasis, which currently only capture skin and ocular morbidity [3].

## 5. Conclusion

This study highlights the alarmingly high prevalence and excess mortality of epilepsy in onchocerciasis-endemic areas of South Sudan. The observed association between ongoing *O. volvulus* transmission and epilepsy, including pNS,

emphasises the need for integrated public health interventions that target both conditions. Strengthening onchocerciasis elimination efforts and ensuring consistent ASM access could markedly reduce epilepsy incidence and mortality in endemic areas. Furthermore, future burden estimates for onchocerciasis should incorporate the impact of OAE or epilepsy on disability and mortality. Taken together, these approaches may mitigate the persistent burden of epilepsy in onchocerciasis-endemic settings.

## Supporting information

**S1 Dataset.  ZIP file containing the 3 datasets used during analysis: 1) Epilepsy suspected and mortality data; 2) Epilepsy confirmed data; 3) anti-Ov16 seroprevalence data per study site.**
(ZIP)

**S1 Appendix.  Contains 1) the household and epilepsy questionnaires used for study (Questionnaires A-E); 2) Statistical comparisons and effect sizes of sociodemographic variables (Tables A and B; Text A); 3) Detailed Anti-Ov16 seroprevalence data per sex, age, county, high/low-risk classification and standardisation (Table C); 4) Suspected epilepsy prevalence per study county and high/low-risk classification (Table D); Positive predictive values of the epilepsy screening question per county and high/low-risk classification (Table E); General interpretation of arcsin-transformed weighted linear regression (Text B).**
(DOCX)

## Acknowledgments

We thank the study villages' residents, persons with epilepsy and their families for their participation in this study. We are also grateful to the clinical officers and research assistants who did the surveys; to Amref Health Africa for using its facilities, equipment and vehicles; and to the Western Equatoria State and Maridi's, Mvolo's, Wulu's, Mundri East and Mundri West Counties governments for their support.

## Author contributions

**Conceptualization:** Stephen Raimon Jada, Joseph N. Siewe Fodjo, Robert Colebunders.

**Data curation:** Luis-Jorge Amaral, Stephen Raimon Jada, Joseph N. Siewe Fodjo, Robert Colebunders.

**Formal analysis:** Luis-Jorge Amaral, Robert Colebunders.

**Funding acquisition:** Luis-Jorge Amaral, Stephen Raimon Jada, Jane Y. Carter, Joseph N. Siewe Fodjo, Robert Colebunders.

**Investigation:** Luis-Jorge Amaral, Stephen Raimon Jada, Jane Y. Carter, Joseph N. Siewe Fodjo, Robert Colebunders.

**Methodology:** Luis-Jorge Amaral, Stephen Raimon Jada, Joseph N. Siewe Fodjo, Robert Colebunders.

**Project administration:** Luis-Jorge Amaral, Stephen Raimon Jada, Jane Y. Carter, Joseph N. Siewe Fodjo, Robert Colebunders.

**Resources:** Stephen Raimon Jada, Yak Yak Bol, Robert Colebunders.

**Software:** Luis-Jorge Amaral.

**Supervision:** Jane Y. Carter, Yak Yak Bol, Joseph N. Siewe Fodjo, Robert Colebunders.

**Validation:** Luis-Jorge Amaral, María-Gloria Basáñez, Charles R. Newton, Robert Colebunders.

**Visualization:** Luis-Jorge Amaral, María-Gloria Basáñez.

**Writing – original draft:** Luis-Jorge Amaral, Robert Colebunders.

**Writing – review & editing:** Luis-Jorge Amaral, Stephen Raimon Jada, Jane Y. Carter, Yak Yak Bol, María-Gloria Basáñez, Charles R. Newton, Joseph N. Siewe Fodjo, Robert Colebunders.

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
