## [Decision Letter · Decision Letter 0]

Please submit your revised manuscript within 30 days, by June 2'nd 2025. If you will need more time than this to complete your revisions, please reply to this message or contact the journal office at plosntds@plos.org. Please include the following items when submitting your revised manuscript:

Response to Reviewers Revised Manuscript with Track Changes Manuscript

Shaden Kamhawi

co-Editor-in-Chief

Paul Brindley

co-Editor-in-Chief

**Additional Editor Comments :**
**Journal Requirements:**

1) Please upload all main figures as separate Figure files in .tif or .eps format. For more information about how to convert and format your figure files please see our guidelines: 

2) We have noticed that you have uploaded Supporting Information files, but you have not included a list of legends. Please add a full list of legends for your Supporting Information files after the references list.

3) Please amend your detailed Financial Disclosure statement. This is published with the article. It must therefore be completed in full sentences and contain the exact wording you wish to be published.

1) State the initials, alongside each funding source, of each author to receive each grant. For example: "This work was supported by the National Institutes of Health (####### to AM; ###### to CJ) and the National Science Foundation (###### to AM)." Please include the initials of the recipients of these grants "40385 and

AID011898."

4) Please ensure that the funders and grant numbers match between the Financial Disclosure field and the Funding Information tab in your submission form. "Note that the funders must be provided in the same order in both places as well. Currently, this grant ((MR/X020258/1) is missing from the Financial Disclosure field.

**Comments to the Authors:**

**Please note that one of the reviews is uploaded as an attachment.**

**Reviewers' comments:**

**Key Review Criteria Required for Acceptance?**

**Methods**

-Are the objectives of the study clearly articulated with a clear testable hypothesis stated?

-Is the study design appropriate to address the stated objectives?

-Is the population clearly described and appropriate for the hypothesis being tested?

-Is the sample size sufficient to ensure adequate power to address the hypothesis being tested?

-Were correct statistical analysis used to support conclusions?

-Are there concerns about ethical or regulatory requirements being met?

Reviewer #1: The objective of the study is clear. This study provides findings that are in line with previous research: epilepsy prevalence is high in onchocerciasis-endemic areas. Findings of increased mortality among PWE compared individuals without epilepsy noted in this study highlights an important public health concern.

Reviewer #2: Thank you for consideration to review an important piece of work.

This is a well designed study with hypothesis stated clearly. Sample size is adequate to provide adequate power of the study.

Clear data analysis plan, and otherwise no

Reviewer #3: -Why do the questionnaires differ for the villages?

-Why did you not ask for the reasons of death in any case? This an important information to draw any conclusion.

-What about previous use of anti-seizure medication? This is also not specified in the questionnaire, but must be taken into account, as well-treated patients are likely to lead a better life and cannot be compared with untreated patients.

- You indicated that: "Given this low coverage and the high OAE and anti-Ov16 prevalence, the South Sudan Ministry of Health increased CDTI frequency to twice a year in 2023 (excepting Wulu, where OAE had not yet been investigated)" but in principle CDTI is not only dependent on OAE but on prevalences of O. volvulus infections.

-Did they also include the age when the seizures started? I did not find this in the questionnaire but this is some kind of criteria to distinguish OAE from other epileptic diseases.

-How was epilepsy actually confirmed? In the table 1 you stated "confirmed epilepsy" but there are no entries, probably this should be inluded in this table and also the information NS positive/nor rather than in second table. Overall it would be helpful to have also the OV16 results given in the table.

**Results**

-Does the analysis presented match the analysis plan?

-Are the results clearly and completely presented?

-Are the figures (Tables, Images) of sufficient quality for clarity?

Reviewer #1: Yes the analyses presented matches the plan and the objective of the study. Figures and tables are clear.

Reviewer #2: Analysis presented match with data analysis plan, results are clearly and completely presented with good quality tables

Reviewer #3: -If 4.6% of pig-rearing were observed, neurocysticercosis should also be taken into account.

-Supplementary table 3 would be easier to understand if children were not pooled due to age and time-point of analysis but rather individual data were given (for each individual the information if seropositive or not and presence of epilepsy/nodding syndrome or not).

-The highest observed seropositivity in Maridi’s high-risk site is is not surprising since people are more exposed in closer proximity to the rivers. And this does not reflect the actual state of O.volvulus infection since most of the people living in the endemic areas had previous contact to the parasite although they do not develop an infection with the filariae as shown in previous literature.

-Looking at the age in the table, does it mean that we have 433 children who were born dead?

- In the discussion you state: "The current study underscores the substantial epilepsy prevalence and mortality among

PWE in onchocerciasis-endemic areas of South Sudan", this sounds again like all cases of epilepsy in onchocerciasis endemic areas are OAE cases but as indicated by the authors in the introduction, epilepsy has several causes: "yet in over 40% of cases, its origin remains unknown". Please make this clearer.

-You describe that in Wulu 4/11 deaths were due to malaria, but according to the table there were more than 30 deaths in this county?

-As indicated above, the conclusion "The statistically significantly lower median age at death for sPWE (20 years) compared to IWE (39 years) further emphasises the premature mortality faced by PWE in these areas" can't be drawn per se, but should be recalculated only for Wulu where the causes of deaths were given.

-You stated that "CDTI coverage across the five counties surveyed was suboptimal, from 24.1% in Mvolo to 66% in Wulu, limiting efforts to control onchocerciasis and OAE", was there a clear drop due to the COVID-19 pandemic?

**Conclusions**

-Are the conclusions supported by the data presented?

-Are the limitations of analysis clearly described?

-Do the authors discuss how these data can be helpful to advance our understanding of the topic under study?

-Is public health relevance addressed?

Reviewer #1: Conclusion and discussion is robust and supported by the analyses and data.

Reviewer #2: Conclusion are well described and supported by the data presented in the study.

A well written study limitations however, it would be good to mention the missed investigations are cost and it's very important to think of applying them in settings where epilepsy is related with onchocerciasis. Yes, it is true that available epidemiological evidence suggest a clear association between epilepsy and onchorceciasis, yet the exact patho-mechanism how onchocerciasis induce epilepsy is unknown. Applying different imaging studies and laboratory test to role out other infections would be of great value.

Yes, authors discuss very well and have shown the gaps to be addressed in the advancement of the topic

Public health relevance is well addressed

Reviewer #3: This study is based on a large amount of collected data, which is really impressive. However, there is a clear weakness with regard to the correct diagnosis of an Ov infection, which is based solely on the detection of antibodies, which would also be positive in healthy people if they had been exposed in the corresponding endemic area. It is therefore questionable whether one can draw the conclusion of an increased mortality linked to a higher prevalence of epilepsy due to an OV infection.

Some limitations were mentioned by the authors, e.g. that the data are based on the questionnaires, the reliability of which is not always given. However, some methodological weaknesses such as the correct diagnosis of OV infection or the lack of information on the cause of death in all villages except Wulu could have been avoided by a better study design (e.g. performance of PCR, palpation of nodules, determination of MF).

**Editorial and Data Presentation Modifications?**

Reviewer #1: Minor Clarification suggested:

Methods: Mortality was ascertained over a wide range of different timeline (18months to 42 months) for each county. Please kindly provide the reason for this and explain if it affected the ascertainment of mortality estimates.

Reviewer #2: No,

Reviewer #3: -Please change the sentence in the introduction: "This gap is particularly concerning in populations with high epilepsy prevalence, such as in highly-endemic onchocerciasis areas,... " since also in areas with low epilepsy prevalence this is a huge concern.

- Please mention in the introduction that several additional causes for epilepsy in O. volvulus endemic areas have been discussed and some studies didn't even find any association with this particular filarial infection and Nodding syndrome (Edrige et al, 2023) and also environmental biotoxins are being discussed to play a pivotal role (Spencer, 2024).

- Table 2 needs formatting, what do the colours (red and yellow) mean?

**Summary and General Comments**

Reviewer #1: Well writing manuscript,

Reviewer #2: Given the relevance of the topic discussed in this article, I would recommend the PLOS NTD to consider the manuscript with Minor Revision. Things to consider;

1. Clearly define key terminologies (e.g Epilepsy and probable nodding syndrome) in the methodology section

2. According to international leagues against epilepsy (ILAE) discourages the use of PWE as people with epilepsy as it shows stigmatization. So it would be of great value to consider use of people with epilepsy rather than a short word (PWE).

3. Paragraph two in introduction section, the author has stated "The epilepsy treatment gap is large in sub-Saharan Africa (SSA), where about 70% of PWE lack adequate care". I think think 70% of people with epilepsy lack anti-seizure medications. It is better the author to be clear

Reviewer #3: As the data are based on an unsound diagnosis of OV infection, any resulting conclusions should be taken with caution.

PLOS authors have the option to publish the peer review history of their article (what does this mean? ). If published, this will include your full peer review and any attached files.

**Do you want your identity to be public for this peer review?** For information about this choice, including consent withdrawal, please see our Privacy Policy .

Reviewer #1: No

Reviewer #2: No

Reviewer #3: No

**Figure resubmission:**
---

## [Editor Report · Decision Letter 1]

Dear DR AMARAL,

We are pleased to inform you that your manuscript 'High epilepsy prevalence and excess mortality in onchocerciasis-endemic counties of South Sudan: A call for integrated interventions' has been provisionally accepted for publication in PLOS Neglected Tropical Diseases.

Best regards,

Angela Monica Ionica, Ph.D.

Academic Editor

Eva Clark

Section Editor

Shaden Kamhawi

co-Editor-in-Chief

Paul Brindley

co-Editor-in-Chief

---

## [Editor Report · Acceptance letter]

Dear Mr Amaral,

We are delighted to inform you that your manuscript, "High epilepsy prevalence and excess mortality in onchocerciasis-endemic counties of South Sudan: A call for integrated interventions," has been formally accepted for publication in PLOS Neglected Tropical Diseases.

Best regards,

Shaden Kamhawi

co-Editor-in-Chief

Paul Brindley

co-Editor-in-Chief
